# Frequency of Asbestos Exposure and Histological Subtype of Ovarian Carcinoma

**DOI:** 10.3390/ijerph19095383

**Published:** 2022-04-28

**Authors:** Pauline Vidican, Olivia Perol, Joëlle Fevotte, Emmanuel Fort, Isabelle Treilleux, Elodie Belladame, Jiri Zavadil, Béatrice Fervers, Barbara Charbotel

**Affiliations:** 1Département Prévention Cancer Environnement, Centre Léon Bérard, 28 Rue Laennec, CEDEX 08, 69373 Lyon, France; olivia.perol@lyon.unicancer.fr (O.P.); elodie.belladame@lyon.unicancer.fr (E.B.); beatrice.fervers@lyon.unicancer.fr (B.F.); 2Inserm UMR1296, “Radiations: Défense, Santé, Environnement”, Centre Léon Bérard, 28 Rue Laennec, 69008 Lyon, France; 3Université de Lyon, Université Lyon 1, Université Gustave Eiffel—Ifsttar, Umrestte, UMR T 9405, Domaine Rockefeller, 8 Avenue Rockefeller, 69008 Lyon, France; joelle.fevotte@fiva.fr (J.F.); emmanuel.fort@univ-lyon1.fr (E.F.); 4Département D’anatomopathologie, Centre Léon Bérard, 28 Rue Laennec, 69008 Lyon, France; isabelle.treilleux@lyon.unicancer.fr; 5Epigenomics and Mechanisms Branch, International Agency for Research on Cancer WHO, 150 Cours Albert Thomas, CEDEX 08, 69372 Lyon, France; zavadilj@iarc.fr; 6Faculté de médecine Lyon Est, Université de Lyon, Université Lyon 1, 8 Avenue Rockefeller, 69008 Lyon, France; 7CRPPE-Lyon, Centre Régional de Pathologies Professionnelles et Environnementales de Lyon, Centre Hospitalier Lyon Sud, Hospices Civils de Lyon, 69495 Pierre Bénite, France

**Keywords:** ovarian cancer, asbestos, occupational exposure, environmental exposure, family exposure

## Abstract

The International Agency for Research on Cancer established a causal link between asbestos exposure and ovarian cancer. However, the exposure frequency and histological characteristics of asbestos-associated ovarian cancers remain to be investigated in detail. This multicenter case–case study assessed the asbestos exposure in ovarian carcinoma (OC) patients, alongside its association with histological subtype. Women were recruited in four hospitals in Lyon, France. Histological reports were reviewed by a pathologist. Patient and family members’ data were collected by phone-based questionnaires. Asbestos exposure was defined as direct (occupational and environmental) and indirect (via parents, partners, and children). An industrial hygienist assessed the probability and level of exposure. The 254 enrolled patients (mean age 60 years) reported having an average of 2.3 different jobs (mean working duration 29 years). The prevalence of direct and indirect asbestos exposure was 13% (mean exposure duration 11 years) and 46%, respectively. High-grade serous carcinoma accounted for 73% of all OCs and 82% of histological subtypes in women with direct exposure. After adjustment on a familial history of OC, no significant associations between asbestos exposure (direct and/or indirect) and high-grade serous carcinoma were found. Women with OC had a high prevalence of asbestos exposure. Establishing risk profiles, as reported here, is important in facilitating compensation for asbestos-related OCs and for the surveillance of women at risk.

## 1. Introduction

Ovarian cancer is the eighth most common cancer and cause of cancer death among women worldwide [1]. The etiology and pathogenesis of ovarian cancer are still not fully understood [2], and overall, less than 10% of all ovarian cancers in France are attributable to established lifestyle factors [3].

At the hormonal level, different factors appear to be associated with ovarian cancer [4]. Nulliparity, early puberty, and late menopause seem to be risk factors [4]. On the other hand, pregnancy (even if not carried to term), multiparity, breastfeeding, especially a prolonged one, would be protective [4,5]. Prolonged use of hormone replacement therapy for menopause would increase the risk [4,6], whereas oral contraceptives would reduce it [4,7]. Studies on the association between infertility, infertility treatments, and risk of ovarian malignancy are still conflicting [8,9]. In addition to reproductive and hormone-related risk factors, family history and genetic factors, tobacco smoking, endometriosis, and body fatness are positively associated with ovarian cancer [10,11,12,13].

Occupational and environmental risk factors for OC remain poorly understood [14]. In 2009, the International Agency for Research on Cancer (IARC) concluded sufficient evidence that asbestos causes ovarian cancer [15]. Reid et al., in their systematic literature review of four cohorts [16], reported statistically significant excess incidence or mortality of ovarian cancer in women exposed to asbestos. Standardized mortality ratios (SMR) in these studies ranged from 4.77 (CI 95% 2.18–9.06) to 2.27 (CI 95% 1.04–4.32), but disease misclassification (with peritoneal mesotheliomas) was noticed. Furthermore, Camargo et al., in their meta-analysis of 18 cohort studies of women occupationally exposed to asbestos [17], reported a pooled SMR for ovarian cancer of 1.77 (CI 95% 1.37–2.28), supporting the IARC conclusion. The authors reported that occupational exposure was more strongly associated with ovarian cancer among cohorts with a SMR for lung cancer >2.0 compared with other cohorts. Using the SMR reported by Camargo et al., IARC estimated that 1.3% of all ovarian cancers in 2015 in France were attributable to occupational asbestos exposure in women [18].

Although the use of asbestos has been banned in numerous industrialized countries, asbestos-related diseases remain a major public health concern. While lung cancer and mesothelioma linked to asbestos received widespread attention, little is known on the prevalence of asbestos exposure in ovarian cancer patients. Previous studies have not investigated histological subtypes of ovarian carcinoma (OC). The aim of this multicenter study was to characterize exposure to asbestos in OC patients and to explore the association with different histological subtypes.

## 2. Materials and Methods

### 2.1. Population and Recruitment

This multicenter case–case study (EVAMOVAIRE) was conducted in four hospitals in Lyon, France, referent for ovarian cancer management. Recruitment was carried out in two phases. Ovarian cancer patients in each center were ascertained by the Medical Information Departments (2010 to 2013) and the weekly multidisciplinary gynecological cancer board (2016 to 2018).

Newly diagnosed (<12 months) French-speaking patients with histologically confirmed primary OC, absence of major deterioration of general health status, and managed in one of the participating centers were eligible for the study. A clinical research assistant checked the women’s general status from the electronic patient record prior to sending an invitation letter and informed consent form to eligible patients. Women who returned their written consent were included in the study. The pathology report for all patients included in the study was systematically reviewed by an expert pathologist to ascertain diagnosis and histologic subtypes. Patients with non-primary epithelial ovarian cancer histologically confirmed by the pathologist were excluded.

### 2.2. Data Collection

After receipt of written consent, patients were contacted by phone to collect the information using four different questionnaires.

A clinician and a clinical research assistant gathered medical data and individual risk factors (medical questionnaire).

Next, a trained investigator (Clininfo Company, Lyon, France) collected the asbestos exposures and completed job histories of the patients and family using standardized questionnaire items for assessing asbestos exposure in epidemiology studies [19,20,21]. Questions including knowledge of asbestos exposure under various circumstances: occupational (asbestos handling, proximity to a colleague working with asbestos, asbestos on premises) and environmental (living in asbestos-flocked premises, present or past residence near a factory that manufactures or uses asbestos, manipulation of asbestos-containing material) were asked (asbestos questionnaire).

The patient’s entire employment history was adapted to female subjects and included items on initial training, employment, and company activities (professional patient curriculum).

To evaluate the possible indirect exposure by an exposed family member (ascendant, partners or descendant), the patient’s family employment history was collected. The year of birth and main occupational activities were asked for each member, focusing on cohabitation periods as well as information on the handling or cleaning of relatives’ work-clothes by the patient (professional patient’s family curriculum).

### 2.3. Assessment of Direct Occupational Exposure

The jobs and sectors were encoded by an industrial hygienist using the International Standard Classification of Occupations of the International Labor Organization (ISCO-68, ILO), the French classification of occupations and Socio-Professional Categories (PCS, INSEE 2003b), and the French Nomenclature of Activities classification (NAF, INSEE 2003a).

The assessment included direct asbestos exposure at work, indirect exposure via nearby colleagues working with asbestos, and occupational environmental exposure. In accordance with epidemiology methods used to characterize past occupational exposure [22,23], the retrospective assessment of asbestos exposure was based on the judgment of an industrial hygienist experienced in this field [19,20,21]. The probability of exposure for each professional occupation was classified as possible (probability of exposure <30%), probable (30–80%), or certain (>80%), according to the technical characteristics of the work, employment dates, degree of certainty of the presence of asbestos in the type of professional occupation, employer, period, and degree of precision of work and activity. As asbestos exposure for female workers tended to be rare and low, any exposure greater than the environmental levels (regulatory environmental threshold 0.005 f/mL) was considered to be as sensitive as possible. The level of exposure was classified as (1) very low to low (more than 0.005 f/mL to less than 30% of the threshold limit value (TLV)); (2) low (30–75% of TLV); and (3) moderate to high (close to or above TLV = 0.1 f/mL).

### 2.4. Assessment of Direct Environmental Exposure

Environmental asbestos exposure from the natural environment, human activity, or domestic activity was estimated using the same method for direct occupational exposure described above. A probability of exposure (possible, probable, certain) and a level of exposure (1, 2, 3) were also assigned.

### 2.5. Assessment of Indirect Exposure via Occupational Exposure of Family Members

Regarding the patients’ occupational exposure, jobs were coded and identified by their activity date. For parents, the date of the beginning of exposure could not be less than the birth date of the subject. For children, the considered dates of employment were between the child’s sixteenth and twenty-fifth birthdays. The exposure for each family member was estimated according to the same principle of probability (possible, probable, or certain) and exposure levels (1, 2, 3) used for direct exposure estimation. For exposure estimation via family, the probability and level of patients were assigned, matching with their most exposed relative. When patients were exposed both directly and indirectly, the preference was given to the direct exposure conclusion.

### 2.6. Statistical Analysis

Frequencies of qualitative variables or means/medians of quantitative variables were calculated using the FREQ or the MEANS procedure, respectively, in the SAS software package, version 9.4. Comparisons of qualitative variables between the first and the second phases of recruitment were carried out with the Χ^2^ test. If the test was not applicable, Fisher’s exact test was performed. Comparisons of quantitative variables between the two groups were made with the Student test. Prevalence ratios (PRs) assessing the relationship between direct and/or indirect exposure to asbestos and high-grade serous carcinoma (versus all other histological subtypes) were estimated by log-binomial regression models using the GENMOD procedure. When the GENMOD procedure did not converge, the SAS COPY macro was used [24].

A total of 270 patients were required to demonstrate a significant association between asbestos exposure and high-grade serous carcinoma (in comparison to other OC subtypes), with a PR of 1.91, an alpha risk of 0.05, and a beta risk of 0.20.

## 3. Results

Overall, 3476 ovarian cancers were ascertained, and 341 out of 568 cases in the first phase (2010–2013) and 330 out of 2908 cases in the second phase (2016–2018) met the inclusion criteria and were invited to participate in the study. In total, 315 returned written informed consent with a response rate of 50% and 44% for the 2010–2013 (171 patients) and 2016–2018 (144 patients) periods, respectively.

Overall, 61 patients were subsequently excluded: 42 patients with non-epithelial ovarian cancer or uncertain histologic diagnosis (after re-examination of the pathology report), 13 cases of secondary ovarian cancer, four women with an incomplete interview, one subsequent refusal to participate, and one double registration (patient recruited by two of the participating hospitals). Finally, 254 patients with OC were included in the EVAMOVAIRE study (Figure 1).

### 3.1. Patients’ Characteristics

The patient’s mean age at the interview was 60.3 ± 11.8 years. The prevalence of established risk and protective factors for OC was as follows: 42% were current or former smokers (17.4 ± 22.4 pack-years), 77% had children (2.2 ± 1 child), and 48% breastfed (7.9 ± 10.4 cumulative months). Oral contraceptive was used by 69% (9.7 ± 7.8 years) and menopausal hormone therapy by 29% (7.2 ± 5.5 years) of patients. In total, 71% had high-grade serous carcinomas (Table 1).

The characteristics of the patients from the two phases of recruitment were comparable in terms of age at diagnosis (1st phase: 59.2 ± 11.9 years, 2nd phase: 61.6 ± 11.6, *p* = 0.1), education level, risk and protective factors for OC, and histological subtype. Patients recruited from 2010 to 2013 had significantly more frequent use of menopausal hormone therapy than those from 2016 to 2018 (33% versus 25%, *p* < 0.0001) (Table 1).

### 3.2. Direct Occupational Exposure

All patients reported at least one occupational period, with a mean of 2.3 ± 1.3 jobs. The mean total working duration was 28.7 ± 11.8 years. According to ILO ISCO, ‘clerical and related workers’ represented 43% of the subjects’ occupations; to PCS, the most represented occupational category was ‘office workers’ with 61% of jobs; to NAF, the most represented occupational sector was ‘wholesale and retail trade, repair of motor vehicles and motorcycles’ with 28% of jobs. Overall, 20% of the patients had blue-collar jobs, skilled or unskilled. The duration of working (1st phase: 28.3 ± 11.7 years, 2nd phase: 28.9 ± 11.5, *p* = 0.6), and sectors and jobs occupied were globally the same between the two groups (Table 2).

In total, 11% and 6% of patients reported that they had been in occupational contact with asbestos and 6% thought they worked in asbestos-flocked premises. The industrial hygienist estimated that 34 patients (13%) had been occupationally exposed to asbestos during 38 occupational periods. The mean duration of asbestos exposure was 10.5 ± 7 years. Prevalence of direct occupational exposure was significantly higher for patients recruited from 2010 to 2013 than from 2016 to 2018 (19% versus 8%, *p* = 0.009) (Table 3).

According to ISCO, ‘service workers’ and ‘production and related workers, transport equipment operators and laborers’ represented 29% each of the occupations of the exposed subjects; to PCS, the most represented occupational category was ‘office workers’ with 38% of jobs; to NAF, the most represented sector was ‘manufacturing’ with 38% of jobs. Overall, 29% of the patients had blue-collar jobs, skilled or unskilled. Sectors and jobs occupied by the exposed subjects were globally the same between the two groups (Table 4).

### 3.3. Direct Environmental Exposure

One patient was considered as having undergone only environmental exposure. This woman lived in Ukraine when she was a child, next to a cement factory, and had no data on occupational exposure to asbestos.

### 3.4. Indirect Family Exposure

For each relative, only the principal occupation during the period of cohabitation was recorded. The family information was collected from one or more children of 69 patients, one or more partners of 224 women as well as mothers and fathers of 146 and 238 patients, respectively. According to ISCO, ‘agricultural, animal husbandry and forestry workers, fishermen and hunters’ represented 25% of the occupations of patients’ mothers; ‘production and related workers, transport equipment operators and laborers’ represented 40%, 39%, and 40% of the occupations of patients’ children, partners, and fathers, respectively.

The industrial hygienist estimated that 117 patients (46%) had been exposed through their families. The relatives of 107 exposed patients (91%) brought their work-clothes home and 92 concerned patients (86%) declared cleaning them. Excluding patients with exposure only (*n* = 15) and both direct and indirect exposure (*n* = 19), 98 patients (45%) had been exposed only through their family members. Prevalence of indirect family exposure was significantly higher in patients from 2010 to 2013 than those from 2016 to 2018 (61% versus 29%, *p* < 0.0001) (Table 5).

### 3.5. Histological Subtype, Risk Factors and Asbestos Exposure

Excluding patients with undetermined-grade serous carcinomas (*n* = 8), high-grade serous carcinoma accounted for 73% of all OCs, 82% of histological subtypes observed in women with direct asbestos exposure, and 72% of histological subtypes observed in women with indirect asbestos exposure.

Univariate analysis suggested significant associations of high-grade serous carcinomas with family history of ovarian cancer (*p* = 0.0005), age (*p* = 0.002), number of children (*p* = 0.005), breastfeeding (*p* = 0.02), difficulties to have children (*p* = 0.07), and tobacco status (*p* = 0.1). In multivariate analysis, only the association with family history of ovarian cancer remained significant (*p* < 0.05).

Concerning asbestos exposure, univariate analysis showed an association between indirect asbestos exposure and high-grade serous carcinomas for a low level of exposure with PR = 1.24, CI 95% 1.04–1.48, *p* = 0.08 (Table 6).

After adjustment on family history of ovarian cancer, no significant association between asbestos exposure (direct and/or indirect) and high-grade serous carcinomas was found (Table 7).

## 4. Discussion

To the best of our knowledge, this is the first study to characterize the prevalence of asbestos exposure in OC patients, by histological subtype, in order to investigate the direct occupational and environmental exposure and indirect exposure via the occupational exposure of family members. Prevalence of direct and indirect asbestos exposure in OC patients was 13% and 46%, respectively. No association with histological subtype was observed. The present study was not designed to estimate the association between asbestos exposure and OC risk, which has been investigated in numerous occupational cohort studies [25].

Current evidence on asbestos exposure and ovarian cancer risk mainly comes from cohorts of occupationally exposed women [26]. The prevalence of occupational exposure to asbestos in our study, assessed as sensitively as possible, appeared to be higher than that observed in women of the same age in the general French population, estimated at 4% [27]. While the characteristics of the patients from the two phases of recruitment were comparable in terms of age at diagnosis, working duration, or sectors and jobs occupied, the prevalence of direct and indirect asbestos exposure was significantly lower in patients from the second phase. This lower prevalence may be explained by the year gap between the two phases (2010 to 2013 and 2016 to 2018). Patients in the first phase had an increased probability to have worked before 1985, a pivotal date in terms of occupational exposure to asbestos in France, as from the 1980s onward, imports of asbestos fell sharply until it was banned in 1997. As a consequence, production of and finished asbestos products and associated exposures decreased accordingly over this period [28,29]. In addition, findings from the Global Burden of Disease Study showed that the decrease in occupational asbestos led to a diminution of the age-standardized death and disability-adjusted life-years rates of ovarian cancer attributable to occupational asbestos exposure from 1990 to 2017 [30]. From the Global Health Data Exchange database [30], the age-standardized disability-adjusted-life-years rates of ovarian cancer attributable to occupational asbestos exposure were 3.42 per 100,000 in 1990 and 2.35 per 100,000 in 2017, which is a great decrease.

Indirect exposure in occupationally unexposed wives and children of asbestos-exposed workers has been shown, since the 1970s, to play a role in the development of asbestos-related cancers, particularly mesothelioma and benign pleural diseases [31,32] as well as ovarian cancer [33]. Radiologic abnormalities were found in the lung in one-third of family members of exposed workers [34]. Inhalation or ingestion of fibers from the hair or clothes of highly exposed workers have been shown to be responsible for domestic exposure, particularly in wives or children cleaning the work-clothes [35,36]. Furthermore, some studies have suggested that the exposure might occur during sexual intercourse with asbestos fibers passing via the genital pathway [37]. In our study, more than 90% of exposed relatives brought their work-clothes home and the large majority of women concerned declared cleaning them. The prevalence of exposure of partners (27%), fathers (24%), and mothers (4%) in our study were lower than the national prevalence (dating from 2010) in the corresponding age groups (i.e., 33% for men over 70 (age group of the patients’ fathers) and 50–70 years (age group of the partners) and 5% for the women 70 (age group of mothers)) [27]. During the recruitment period of our study between 2010 and 2018, we can assume that the prevalence of asbestos exposure had decreased. In addition, the questions were addressed to the patients, and not directly to their relatives and may have caused some loss information on occupations, especially the oldest ones, hence the related prevalence could be underestimated.

The biological plausibility of ovarian cancer derives in part from asbestos fibers in exposed women’s ovaries [38]. Some studies [37] have reported up to twice the amount of asbestos fibers in the ovarian tissue of women with exposed family members compared to women without exposure. However, the relationship between asbestos exposure and ovarian cancer is not understood as well as other asbestos-related diseases. The distribution of fibers throughout the body after inhalation remains uncertain. The fibers could migrate through the diaphragm and the peritoneal cavity. They could also penetrate the epithelial and interstitial cells, access the lymphatic system, and reach the peritoneum and the peritoneal cavity [16,25]. The asbestos fibers would then penetrate and accumulate (without being able to be excreted) in the ovarian tissues, leading to persistent local inflammation at the origin of tissue lesions and genetic and epigenetic alterations, which may, in turn, promote ovarian tumorigenesis [25,39]. Current evidence supports the role of inflammation in the development of ovarian cancer [40,41].

No significant association was found with histological subtype, suggesting that asbestos exposure does not cause a particular histological subtype of OC. However, this study has possible limitations with regard to conclusive findings. First, the response rate (47%) was only moderate, even though it was comparable to other studies in this patient population with a poor prognosis disease. People identified most often as exposed to asbestos belong to low socio-professional categories and the participation rate is generally lower in these categories. Although the study was presented to the participants as aiming to understand the environmental risk factors, and not specifically asbestos, and the association between exposure to asbestos and ovarian cancer was largely unknown at the time of recruitment (by both the general population and clinicians), recruitment bias cannot be completely excluded. Next, 204 patients were not eligible due to criteria involving death, deterioration of the general condition, or relapse. This might have caused a selection bias as they could have cancers with a worse prognosis, and exposure to asbestos could be linked to more aggressive forms of cancer [42]. Conversely, 472 patients had a diagnosis of more than one year and 46 were managed externally, so they could have cancers with a better prognosis. Moreover, as misclassification of peritoneal mesotheliomas has been suggested in the literature [16], all pathological reports of included patients were retrieved and reviewed by the same pathologist (specialist in ovarian cancers) and 27 patients had to be excluded to avoid histological misclassification. Thus, while we estimated that 270 subjects needed to show a PR of 1.91, only 246 patients were included in this part of the analysis. With 246 patients, an alpha risk of 5%, a beta risk of 20%, and an exposure prevalence of 15.6% in patients with high-grade serous carcinomas, the significantly observable prevalence ratio would have been at least 2.1. We may lack statistical power to show that asbestos exposure increases the risk of developing high-grade serous carcinoma. Finally, another explanation can be that our analysis of indirect exposure was too sensitive. Level of patients were assigned matched with their most exposed relative, so perhaps only a certain level of exposure in relatives (level 2 or 3) generated relevant exposure in the index case.

Except in rare cohort studies, exposure assessment is generally retrospective in occupational epidemiology. In this case, the best way to perform a relevant and unbiased assessment is to perform an expert assessment with an industrial hygienist blinded to the histological subtypes [22,23]. Due to the great latency between exposure and cancer occurrence, asbestos fibers may have disappeared. Consequently, the absence of asbestos fibers does not mean the absence of exposure. Moreover, to date, standardized methods for the determination of asbestos fibers in tissue samples have not been routinely available and heterogeneity in sample conservation (fresh frozen and formalin-fixed paraffin-embedded tissue samples) does not allow for reliable assessments. Information on other asbestos-related diseases such as plaques, pleurisy, or fibrosis have not been collected. However, while these signs may provide arguments for past exposure, they are rare, even among highly exposed subjects [43,44]. Finally, reliable environmental and geographic information on asbestos (from the natural environment or asbestos processing facilities) are not available in France, and could therefore not be assessed in the present study.

While the prevalence of exposure close to or above the TLV in effect (0.1 f/mL) was similarly low (<1%) compared to national data [27], we cannot exclude that the exposure assessment in the present study had been more sensitive compared to the job exposure matrix used in the national study [45]. Moreover, using mean exposure values over a given work period, as in our study, may have led to smoothing out peak exposures: strong but rare exposures resulted in low mean exposure values. Furthermore, our study did not consider common exposures to asbestos from domestic appliances (e.g., ironing, kitchen stove, etc.) and building materials (e.g., garage roof, etc.) as at that given time point in France, most French women might have been exposed.

## 5. Conclusions

This study, with a detailed investigation of all types of exposure (occupational, environmental, and family-based), showed that women with OC have a high prevalence of asbestos exposure, although it is difficult to compare the results to other existing prevalence data due to the lack of similar studies. The epithelial histological type representing 90% of invasive ovarian cancers, these findings provide an additional argument to classify asbestos as a proven ovarian carcinogen [25]. Concerning the hypothesis that exposure to asbestos could favor the development of a particularly aggressive histological subtype of OC (high-grade serous carcinomas), we did not find any significant association. It was either due to lack of statistical power or because asbestos might not be a specific risk factor for any histological subtype of OC. Further studies are needed in this regard including analyses of comprehensive molecular profiles in the tumors of distinct histological subtypes derived from asbestos-exposed and unexposed OC patients. Our results provide useful information for clinicians to pay more attention to asbestos exposure in patients with ovarian cancer. Finally, refining the knowledge to establish risk profiles seems essential to (i) identify a group of women at risk of developing OC due to their occupational or non-occupational exposure to asbestos, and provide them with appropriate surveillance to detect OC at an early stage, and (ii) to give women the opportunity to benefit from recognition and compensation for asbestos-related OCs [46].

## Figures and Tables

**Figure 1 ijerph-19-05383-f001:**
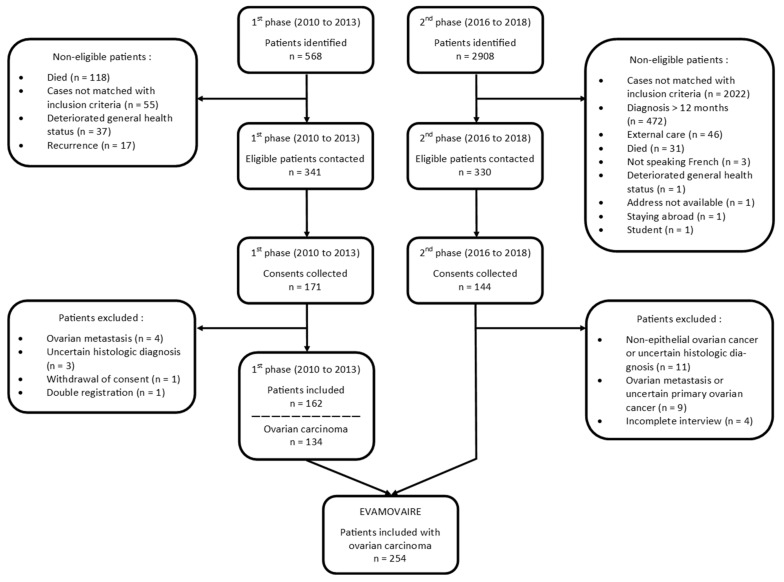
Flowchart.

**Table 1 ijerph-19-05383-t001:** Patient characteristics.

Patients’ Characteristics	EVAMOVAIRE *N* = 254 *n* (%)	1st Phase *N* = 134 *n* (%)	2nd Phase *N* = 120 *n* (%)	*p*-Value
Education				0.2
None, primary school certificate, or technical secondary school certificate	115 (45.3)	66 (49.3)	49 (40.8)	
Bachelor’s degree or higher education	139 (54.7)	68 (50.7)	71 (59.2)	
Current or former smoker	107 (42.1)	64 (47.8)	43 (35.8)	0.06
Body mass index (maximum weight in life)				0.9
<25	130 (51.2)	68 (50.8)	62 (51.7)	
(25–30)	78 (30.7)	42 (31.3)	36 (30.0)	
≥30	46 (18.1)	24 (17.9)	22 (18.3)	
Number of children				0.02
0	59 (23.2)	28 (20.9)	31 (25.8)	
1	39 (15.4)	16 (11.9)	23 (19.2)	
2	93 (36.6)	61 (45.5)	32 (26.7)	
3 and more	63 (24.8)	29 (21.6)	34 (28.3)	
Breastfeeding	123 (49.0)	65 (48.5)	58 (49.6)	0.9
Difficulties to have children	43 (16.9)	19 (14.2)	24 (20.0)	0.2
Infertility treatments	22 (8.7)	10 (52.6)	11 (45.8)	0.7
Use of oral contraceptive	175 (68.9)	90 (67.2)	85 (70.8)	0.5
Use of hormone replacement therapy for menopause	74 (29.3)	44 (32.8)	30 (25.2)	0.06
Family history of ovarian cancer	30 (11.8)	11 (8.2)	19 (15.8)	<0.0001
Histological subtype				0.3 *
Serous high-grade	180 (70.9)	90 (67.2)	90 (75.0)	
Serous low-grade	14 (5.5)	8 (6.0)	6 (5.0)	
Serous undetermined-grade	8 (3.1)	6 (4.5)	2 (1.7)	
Endometrioid	16 (6.3)	11 (8.2)	5 (4.2)	
Clear cells	17 (6.7)	8 (6.0)	9 (7.5)	
Mucinous	11 (4.3)	7 (5.2)	4 (3.3)	
Brenner’s	2 (0.8)	1 (0.7)	1 (0.8)	
Mixed	2 (0.8)	0 (0)	2 (1.7)	
Undifferentiated	1 (0.4)	0 (0)	1 (0.8)	
Others	3 (1.2)	3 (2.2)	0 (0)	

* Fisher’s exact test.

**Table 2 ijerph-19-05383-t002:** Distribution of patients’ jobs by ISCO, PCS, and NAF.

Patients’ Jobs	EVAMOVAIRE *N* = 254 *n* (%)	1st Phase *N* = 134 *n* (%)	2nd Phase *N* = 120 *n* (%)
ISCO Level 1			
Scientific, professional, technical and related workers (0/1)	83 (32.7)	41 (30.6)	42 (35.0)
Administrative and managerial workers (2)	5 (2.0)	1 (0.7)	4 (3.3)
Clerical and related workers (3)	110 (43.3)	58 (43.3)	52 (43.3)
Sales workers (4)	55 (21.7)	32 (23.9)	23 (19.2)
Service workers (5)	68 (26.8)	40 (29.9)	28 (23.3)
Agricultural, animal husbandry and forestry workers, fishermen and hunters (6)	11 (4.3)	8 (6.0)	3 (2.5)
Production and related workers, transport equipment operators and laborers (7/8/9)	43 (16.9)	25 (18.7)	18 (15.0)
Total *	375	205	170
PCS Level 1			
Farmers (1)	8 (3.1)	6 (4.5)	2 (1.7)
Self-employed, shopkeepers or chief executive officers (2)	17 (6.7)	9 (6.7)	8 (6.7)
Executive or higher intellectual professions (3)	34 (13.4)	12 (9.0)	22 (18.3)
Intermediate professions (4)	95 (37.4)	49 (36.6)	46 (38.3)
Office workers (5)	156 (61.4)	89 (66.4)	67 (55.8)
Manual workers (6)	50 (19.7)	30 (22.4)	20 (16.7)
Total *	360	195	165
NAF			
Agriculture, hunting, forestry (A)	15 (5.9)	10 (7.5)	5 (4.2)
Fishing, aquaculture, related service (B)	0 (0)	0 (0)	0 (0)
Mining and quarrying (C)	0 (0)	0 (0)	0 (0)
Manufacturing (D)	67 (26.4)	38 (28.4)	29 (24.2)
Electricity, gas, water supply (E)	3 (1.2)	2 (1.5)	1 (0.8)
Construction (F)	6 (2.4)	2 (1.5)	4 (3.3)
Wholesale and retail trade, repair of motor vehicles and motorcycles (G)	71 (28.0)	41 (30.6)	30 (25.0)
Accommodation and food service activities (H)	22 (8.7)	16 (11.9)	6 (5.0)
Transport, storage and communications (I)	16 (6.3)	11 (8.2)	5 (4.2)
Financial activities (J)	19 (7.5)	11 (8.2)	8 (6.7)
Real estate, renting, and business activities (K)	39 (15.4)	18 (13.4)	21 (17.5)
Public administration (L)	33 (13.0)	16 (11.9)	17 (14.2)
Education (M)	49 (19.3)	23 (17.2)	26 (21.7)
Health and social work (N)	64 (25.2)	33 (24.6)	31 (25.8)
Other community, social and personal services activities (O)	23 (9.1)	11 (8.2)	12 (10.0)
Activities of private households as employers and undifferentiated production activities of private households (P)	26 (10.2)	17 (12.7)	9 (7.5)
Extraterritorial organizations and bodies (Q)	0 (0)	0 (0)	0 (0)
Total *	453	249	204

* The sum of the frequencies is greater than 100% since the patients could have worked in several types of professions/sectors during their working life. ISCO: International Standard Classification of Occupations 1968. PCS: French classification of occupations and Socio-Professional Categories 2003. NAF: French Nomenclature of Activities classification 2003.

**Table 3 ijerph-19-05383-t003:** Estimation of patients’ occupational exposure.

Patients’ Occupational Exposure	EVAMOVAIRE *N* = 254 n (%)	1st Phase *N* = 134 n (%)	2nd Phase *N* = 120 n (%)	*p*-Value
Asbestos occupational exposure				0.009
No	220 (86.6)	109 (81.3)	111 (92.5)	
Yes	34 (13.4)	25 (18.7)	9 (7.5)	
Probability of exposure				0.02 *
Possible	14 (5.5)	8 (6.0)	6 (5.0)	
Probable	12 (4.7)	10 (7.5)	2 (1.7)	
Certain	8 (3.2)	7 (5.2)	1 (0.8)	
Exposure level				0.04 *
Level 1	28 (11.0)	20 (14.9)	8 (6.7)	
Level 2	5 (2.0)	4 (3.0)	1 (0.8)	
Level 3	1 (0.4)	1 (0.8)	0 (0)	

* Fisher’s exact test: Level 1: very low to low (more than 0.005f/mL to less than 30% of threshold limit value (TLV)); Level 2: low (30–75% of TLV); Level 3: moderate to high (close to or above TLV = 0.1 f/mL).

**Table 4 ijerph-19-05383-t004:** Jobs of exposed patients by ISCO, PCS, and NAF.

Jobs of Exposed Patient	EVAMOVAIRE *N* = 254 n (%)	1st Phase *N* = 134 n (%)	2nd Phase *N* = 120 n (%)
ISCO Level 1			
Scientific, professional, technical, and related workers (0/1)	7 (20.6)	5 (20.0)	2 (22.0)
Clerical and related workers (3)	6 (17.6)	4 (16.0)	2 (22.2)
Sales workers (4)	2 (5.9)	1 (4.0)	1 (11.1)
Service workers (5)	10 (29.4)	9 (36.0)	1 (11.1)
Production and related workers, transport equipment operators and laborers (7/8/9)	10 (29.4)	7 (28.0)	3 (33.3)
Total *	35	26	9
PCS Level 1			
Executive or higher intellectual professions (3)	2 (5.9)	2 (8.0)	0 (0)
Intermediate professions (4)	9 (26.5)	7 (28.0)	2 (22.2)
Office workers (5)	13 (38.2)	10 (40.0)	3 (33.3)
Manual workers (6)	10 (29.4)	6 (24.0)	4 (44.4)
Total *	34	25	9
NAF			
Manufacturing (D)	13 (38.2)	10 (40.0)	3 (33.3)
Construction (F)	2 (5.9)	0 (0)	2 (22.2)
Wholesale and retail trade, repair of motor vehicles and motorcycles (G)	4 (11.8)	3 (12.0)	1 (11.1)
Accommodation and food service activities (H)	1 (2.9)	1 (4.0)	0 (0)
Transport, storage and communications (I)	1 (2.9)	0 (0)	1 (11.1)
Real estate, renting and business activities (K)	3 (8.8)	3 (12.0)	0 (0)
Public administration (L)	1 (2.9)	1 (4.0)	0 (0)
Education (M)	5 (14.7)	3 (12.0)	2 (22.2)
Health and social work (N)	2 (5.9)	2 (8.0)	0 (0)
Other community, social and personal services activities (O)	4 (11.8)	4 (16.0)	0 (0)
Total *	36	27	9

* The sum of the frequencies is greater than 100% since the patients could have worked in several types of professions/sectors during their working life. ISCO: International Standard Classification of Occupations 1968. PCS: French classification of occupations and Socio-Professional Categories 2003. NAF: French Nomenclature of Activities classification 2003.

**Table 5 ijerph-19-05383-t005:** Estimation of patient exposure only through their family (*N* = 219).

Patients’ Exposure only through Their Family	EVAMOVAIRE *N* = 219 * n (%)	1st Phase *N* = 108 * n (%)	2nd Phase *N* = 111 * n (%)	*p*-Value
Asbestos occupational exposure				<0.0001
No	121 (55.3)	42 (38.9)	79 (71.2)	
Yes	98 (44.7)	66 (61.1)	32 (28.8)	
Probability of exposure				<0.0001
Possible	12 (5.5)	7 (6.5)	5 (4.5)	
Probable	25 (11.4)	14 (13.0)	11 (9.9)	
Certain	61 (27.9)	45 (41.7)	16 (14.4)	
Exposure level				<0.0001
Level 1	64 (29.2)	45 (41.7)	19 (17.1)	
Level 2	23 (10.5)	15 (13.9)	8 (7.2)	
Level 3	11 (5.0)	6 (5.6)	5 (4.5)	

* Patients with direct exposure only and both direct and indirect exposure were excluded. Level 1: very low to low (more than 0.005 f/mL to less than 30% of threshold limit value (TLV)). Level 2: low (30–75% of TLV). Level 3: moderate to high (close to or above TLV = 0.1 f/mL).

**Table 6 ijerph-19-05383-t006:** Asbestos exposure and histological subtype, univariate analysis (*N* = 246).

Asbestos Exposure	Total * *n* (%)	High-Grade Serous Carcinomas *n* (%)	Another Histological Subtype *n* (%)	PR	CI 95%	*p*-Value
Direct exposure to asbestos						0.2
No	212 (86.2)	152 (84.4)	60 (90.9)	1	-	
Yes	34 (13.8)	28 (15.6)	6 (9.1)	1.15	0.96–1.37	
Probability of exposure						0.6
Not exposed	212 (86.2)	152 (84.4)	60 (90.9)	1	-	
Possible	15 (6.1)	12 (6.7)	3 (4.6)	1.11	0.85–1.46	
Probable	11 (4.5)	9 (5.0)	2 (3.0)	1.14	0.85–1.53	
Certain	8 (3.2)	7 (3.9)	1 (1.5)	1.22	0.93–1.61	
Exposure level						0.4
Not exposed	212 (86.2)	152 (84.4)	60 (90.9)	1	-	
Level 1	28 (11.4)	23 (12.8)	5 (7.6)	1.15	0.95–1.39	
Level 2 or Level 3 **	6 (2.4)	5 (2.8)	1 (1.5)	1.16	0.80–1.68	
Indirect exposure only						0.6
No	119 (56.1)	87 (57.2)	32 (53.3)	1	-	
Yes	93 (43.9)	65 (42.8)	28 (46.7)	0.96	0.80–1.14	
Probability of exposure						0.5
Not exposed	119 (56.1)	87 (57.2)	32 (53.3)	1	-	
Possible	12 (5.7)	8 (5.3)	4 (6.7)	0.91	0.60–1.38	
Probable	25 (11.8)	15 (9.9)	10 (16.7)	0.82	0.58–1.15	
Certain	56 (26.4)	42 (27.6)	14 (23.3)	1.03	0.85–1.24	
Exposure level						0.08
Not exposed	119 (56.1)	87 (57.2)	32 (53.3)	1	-	
Level 1	62 (29.2)	39 (25.7)	23 (38.3)	0.86	0.69–1.07	
Level 2	21 (9.9)	19 (12.5)	2 (3.3)	1.24	1.04–1.48	
Level 3	10 (4.7)	7 (4.6)	3 (5.0)	0.96	0.63–1.46	
Direct and indirect exposure						0.5
Not exposed	119 (48.4)	87 (48.3)	32 (48.5)	1	-	
Indirect exposure only	93 (37.8)	65 (36.1)	28 (42.4)	0.96	0.80–1.14	
Direct exposure only	15 (6.1)	12 (6.7)	3 (4.6)	1.09	0.83–1.44	
Direct and indirect exposure	19 (7.7)	16 (8.9)	3 (4.6)	1.15	0.92–1.44	

* Serous carcinomas with undetermined-grade were excluded. ** No patients with level 3 direct exposure and another histological subtype, levels 2 and 3 were combined.

**Table 7 ijerph-19-05383-t007:** Asbestos exposure and histological subtype, multivariate analysis (*N* = 246).

Asbestos Exposure	PR	CI 95%	*p*-Value
Direct exposure to asbestos			0.3
No	1	-	
Yes	0.95	0.85–1.05	
Probability of exposure			0.3
Not exposed	1	-	
Possible	0.9	0.73–1.10	
Probable	1	0.97–1.03	
Certain	1.25	0.95–1.65	
Exposure level			0.6
Not exposed	1	-	
Level 1	0.93	0.83–1.06	
Level 2 or Level 3 *	1	0.97–1.04	
Indirect exposure only			0.3
No	1	-	
Yes	0.94	0.86–1.04	
Probability of exposure			0.4
Not exposed	1	-	
Possible	0.92	0.61–1.39	
Probable	0.81	0.63–1.05	
Certain	1	0.97–1.03	
Exposure level			0.4
Not exposed	1	-	
Level 1	0.86	0.73–1.02	
Level 2	1	0.95–1.06	
Level 3	0.95	0.63–1.44	
Direct and indirect exposure			0.8
Not exposed	1	-	
Indirect exposure only	0.98	0.83–1.16	
Direct exposure only	1	0.97–1.03	
Direct and indirect exposure	0.92	0.79–1.08	

* No patients with level 3 direct exposure and another histological subtype, levels 2 and 3 were combined.

## Data Availability

The data presented in this study are available on request from the corresponding author.

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
