# Peer review of "Frequency of Asbestos Exposure and Histological Subtype of Ovarian Carcinoma"

_ijerph, 2022, doi:10.3390/ijerph19095383_

Round 1

Reviewer 1 Report

I very much appreciated the work of the authors who addressed the interesting issue of asbestos exposure in a group of women with ovarian cancer.

I have only few questions/suggestions:

1) Some women were not able to participate to the study due thei bad health conditions. Could you define more in detail which were clinical conditions that prevent these women to participate?

2) The Authors correctly underlined a low participation rate (about 50% or lower). Are there any significant differences between women enrolled and not enrolled for the study? I mean, are there any difference considering age, or other variables known by the administrative dataset used to select patients to be caontacted? These differences are significant? Taking into account these possible differrences may help in discussing results.

3) Was any clinical sign of asbestos exposure available? For example, some of the enrolled women had some chest radiography that could had documented begnin pleural lesions? If yes, it could be a group with documented exposure (maybe at higher levels)  that could have different features from the others, and could provide further information on the topic.

Reviewer 2 Report

Authors present a detailed study on the correlation between asbestos exposure and ovarian cancer. Manuscript is well written and data presented have potentially useful clinical application.

I have only a few suggestions to improve the overall quality:

  • Molecular profiles (especially serous) of ovarian cancer (p53 mutations) should be included in the discussion and related to DNA damage induced by asbestos
  • Geographic details on asbestos exposure in the studied cohort are missing: State, region town ecc..(it would be useful to include this information in the results or tables)
  • Based on the geographic location of exposure, other asbestos-like fibers such as erionite and fluoro-edenite may be detected in the environment. Therefore, authors should state if all patients have been exposed to asbestos fibers vs other fibres.

Reviewer 3 Report

The topic is of interest but the data are  insufficient to address reliably the hypotheses. Overall this is a very small cohort and the authors conclusions are based on estimates of whether it is likely or not that the subjects studied were exposed to asbestos.  Reconstructions of exposure made years after this has occurred are of course estimates and as all estimates have limited scientific accuracy.  In summary, in the present status these data are preliminary, and need to be validated.

The Authors could try to validate their findings with some objective evidence of exposure, for example a review of the imaging of the subjects studied to validate evidence of exposure or lack of exposure. Similarly if autopsies were performed lung content analyses should be performed. Or at least insubjects in which  surgeries were performed the presence and type of asbestos in the abdominal lymph nodes or even in the ovaries should be studied.  Alternatively, if for some reason the authors have no access to imaging and if lung content analyses were not performed, lymph nodes are not accessible, etc., and thus everything is based on hypothetical exposure reconstruction, then at least the same study should be performed in a separate cohort to validate these hypothetical findings  independently.

Also some hard mineralogical evidence of the type of asbestos exposure in these women and whther it matches the work history or indirect exposure would be helpful.

Also the authors discuss the finding that peritoneal inflammation favors the development ovarian carcinoma. Asbestos is one of the causes of peritoneal inflammation.  The authors should study also the previous history of abdominal/surgical surgeries in their cohort to see whether they can separate this cause of peritoneal inflammation to the possible exposure of these women to asbestos. 

Reviewer 4 Report

The Authors study the correlation between asbestos exposure and ovarian cancers. The research is detailed and complete with scientific strictness. Statistical limitation is due to recruitment method of cases: the voluntary response to a survey, expecially in case of <50% of adhesion, is of low reliability. This topic is well present in discussion session (330-357).

In a logic principle, but it is only my opinion shared by others, the term "histologic subtypes of ovarian cancer" is not correct because it defines very different ovarian diseases. I should prefer the term "histologic types"
